# "I'm tired of black boxes!": A systematic comparison of faculty well-being and need satisfaction before and during the COVID-19 crisis

Carolin Schwab[1]*, Anne C. Frenzel[1], Martin Daumiller[2], Markus Dresel[2], Oliver Dickhäuser[3], Stefan Janke[3], Anton K. G. Marx[1]

1 Department of Psychology, University of Munich, Munich, Germany, 2 Department of Psychology, University of Augsburg, Augsburg, Germany, 3 Department of Psychology, University of Mannheim, Mannheim Germany

* carolin.schwab@psy.lmu.de

**Data Availability Statement:** All relevant data are available in OSF at https://osf.io/b9y4a/.

## Abstract

As of today, surprisingly little is known about the subjective well-being of faculty in general, but especially when teaching online and during a time of pandemic during lockdowns in particular. To narrow this research gap, the present study systematically compared the subjective well-being of faculty teaching face-to-face before to those teaching online during the COVID-19 pandemic, adopting a self-determination theory framework. The data reported here stem from a study conducted before the pandemic (Sample 1, $n = 101$) and which repeated-measures survey design we replicated to collect corresponding data during the pandemic (Sample 2, $n = 71$). Results showed that faculty teaching online during the pandemic reported impaired satisfaction of all three basic needs, that is reduced autonomy, competence, and especially relatedness, as well as impaired subjective well-being (clearly reduced enjoyment and reduced teaching satisfaction; increased anger and a tendency towards more shame) compared to faculty teaching face-to-face before the pandemic. Yet pride, anxiety, and boredom were experienced to a similar extent across both samples. The effects of the teaching format on the different aspects of subjective well-being were overall mediated in self-determination-theory-congruent ways by the satisfaction of the basic needs for autonomy, competence, and relatedness. We conclude for a post-pandemic future that online teaching will supplement rather than replace face-to-face teaching in higher education institutions, as their importance for building relationships and satisfying social interactions not only for students but also for faculty seem to have been underestimated so far.

## Introduction

Excerpt from a Twitter thread:

"God bless the students who nod enthusiastically during a lecture" [1]

"And those on Zoom, who turn on their cameras" [2]

**Funding:** The author(s) received no specific funding for this work.

**Competing interests:** The authors have declared that no competing interests exist.

"Those are heros [sic]!" [3]

The onset of the COVID-19 pandemic in early 2020 implying public lock-downs aimed at mitigating the spread of the new virus inevitably also hit universities worldwide. They had to close down lecture halls and very suddenly stop the still predominant face-to-face teaching [4,5]. In striving to keep up teaching and learning in higher education, countries worldwide tried to move to online teaching as quickly as possible and many aimed for video-based digital tools to allow the offering of synchronous online teaching formats [6–8]. While this forced shift boosted digitalization in the higher education context, it imposed great challenges not only for students who needed to demonstrate very high self-managerial and self-directed learning skills to master digital learning [9] but also for faculty, that is all individuals teaching at a higher education institution irrespective of the held degree or exact position, thereby also including for instance doctoral students and external lecturers. All faculty needed to quickly re-think and adapt their established ways of teaching to transfer their classes to online environments [10].

Given the acknowledged importance of emotions in higher education [11], as of today, surprisingly little is known about the subjective well-being of faculty, especially when teaching online [12,13]. To narrow this research gap, the present study aimed to compare the subjective well-being of faculty teaching synchronous online classes using an online meeting tool after the COVID-19 crisis hit, against those teaching face-to-face classes before the pandemic. Thereby, this study contributes to understanding how a worldwide stressor, such as a pandemic, affects teaching in higher education and provides information on the experiences of faculty during pandemic-enforced emergency online teaching. These insights may help to deal with and prepare for such stressors in the future and shed some light onto the factors and mechanisms that contribute to faculty well-being in online teaching in a post-pandemic era more generally.

By online teaching we refer to purely digital online offers that do not include any on-site in-person meetings and which may take place synchronously, that is faculty and students meet at the same time, or asynchronously, that is provision of learning opportunities that are worked on at different times; by face-to-face teaching we refer to purely on-site in-person classroom teaching. We conceptualized well-being as the combination of frequent positive emotions, infrequent negative emotions, and high teaching satisfaction [14] and adopted self-determination theory (SDT; [15]) as theoretical framework.

## SDT and faculty well-being in face-to-face teaching

Self-determination theory proposes that an individual's motivation and well-being depend on the satisfaction of the three basic psychological needs for autonomy, competence, and relatedness. Individuals experience autonomy when they have agency over their actions, experience competence when they successfully use their skills to interact with their environment, and experience relatedness when they sense a connection and mutual caring with others [15]. Ample research in diverse settings and populations showed that the satisfaction of the three basic needs did not only positively predict autonomous forms of motivation [16] and general subjective well-being [17], but also further outcomes, such as achievement, engagement, satisfaction (e.g., [18,19]), and positive emotional experiences [20–23]. Conversely, a lack of experiencing autonomy, competence, and relatedness has been shown to be linked with increased negative emotional experiences (e.g., [20,22,23]). Taken together, need satisfaction seems to be associated with well-being in various settings.

Prior research supports those postulated associations between need satisfaction and subjective well-being, including emotional experiences and satisfaction, also for teaching professions.

Research on teachers at various school types showed that autonomy, competence, and relatedness were positively associated with positive affect and enjoyment, and negatively with negative affect and negative emotions, such as anxiety and anger, during teaching [20,22,24,25]. Research on the emotional experiences of faculty is comparably scarce [26], although faculty do experience a variety of different emotions during their workdays [27]. Nevertheless, there is initial evidence from the higher education context that the satisfaction of one or multiple psychological needs was associated with intrinsic motivation, higher levels of positive emotions, lower levels of negative emotions, and higher (teaching) satisfaction [16,18,28–32]. Taken together, the associations between need satisfaction and well-being seem to apply to teaching contexts in both schools and higher education. This research, however, is limited to face-to-face teaching—empirical studies on such explicit associations during online teaching are still lacking (as noted for instance by [12,13]).

## Faculty need satisfaction in response to the COVID-19 crisis onset and associations with well-being in emergency online teaching

At the onset of the COVID-19 pandemic, in many western, well-developed countries including Germany where the present study was conducted, video-based digital tools were quickly available and allowed to offer synchronous online teaching formats. Such formats, however, can be compromised by problems such as bandwidth, poor audio quality, and complex technological handling [10,33]. Below we review existing research exploring faculty experiences in different teaching contexts and deliberate about the exceptional circumstances with respect to emergency online teaching during the COVID-19 crisis in order to derive hypotheses regarding how faculty members' basic need satisfaction and subjective well-being might have been affected. Generally, it needs to be considered that the COVID-19 crisis and associated changes in the workplace severely threatened the psychological needs of individuals, even if faculty may not have been immediately at risk to lose their jobs [34,35].

Regarding the satisfaction of the need for autonomy, to the degree that faculty experienced a lack of freedom in determining the content, activities, or policies in class, their perceived autonomy could be thwarted, as has been shown for graduate teaching assistants in face-to-face teaching [36]. During the onset of the COVID-19 crisis, on the one hand, the rapid shift from familiar face-to-face to unfamiliar online teaching may have impaired faculty members' perceived autonomy because the new format may not have aligned very well with their ideas about teaching, may have made their habitually used face-to-face teaching methods inapplicable, and may have imposed the challenge to develop new teaching methods, thus possibly reducing autonomy within teaching sessions (e.g., [37]). On the other hand, due to the unprecedented circumstances, faculty were typically offered maximum flexibility when it came to maintaining their teaching activities and could decide how to offer their classes with sample options ranging from interactive, synchronous sessions using an online meeting tool to asynchronous, purely text-based self-learning units (e.g., [10]). This choice likely even increased their freedom with respect to workplace and time management because online courses do not require physical presence at university and asynchronous offers do not even require attendance at a specific time. Assuming that faculty presumably chose the online teaching approach that fit their own preferences, teaching conceptions, and competencies best (e.g., [38]) and that they gained freedom with respect to workplace and time during their work days ([39]; analogous to findings in a student sample; see [40]), their perceived autonomy should not have been impaired because the transition to emergency online teaching allowed for new forms of control and agency on a general level. Overall, considering that faculty may have lost but also gained some autonomy during the onset of the COVID-19 crisis, we had no reason to assume

a reduced level of satisfaction of the need for autonomy among faculty teaching during the COVID-19 crisis as compared to those teaching before.

Regarding the satisfaction of the need for competence, most faculty lacked experience and training in online teaching (e.g., [8,37,41]), thus faculty may have experienced a lack of the pedagogical and technical skills to implement effective online teaching that aligned with their beliefs about good teaching [42], resulting in lowered subjective competence. With respect to online teaching, the concept of competence may be conceived broader than in face-to-face teaching because faculty not only needed to master the manifold teaching task itself and acquire new didactical concepts to teach content in online settings, but also master the new digital tools and acquire technical methods to implement online teaching in the first place [10,43]. In prior research, faculty reported technical problems and issues with engaging students in class discussions as challenges in online teaching [44–46]. Based on the assumption that both pedagogical and technical skills influence perceived competence to teach online [42], it is likely that faculty members' perceived competence was lower among those faculty teaching during the onset of the COVID-19 crisis compared against those teaching before. Moreover, it seems reasonable that faculty perceived their competence within emergency only teaching settings to be lower, when more technical problems occurred.

Regarding the satisfaction of the need for relatedness, we expected the physical distance between faculty and their students to be the most salient negative factor when transitioning to COVID-19-enforced emergency online teaching. Whereas content may have been covered in similarly effective ways especially in lectures [47], many of the interactions in which faculty and students engaged in before, during, and after class in face-to-face settings may not have been resembled sufficiently in online environments, thus undermining opportunities to build relationships and experience a sense of belonging and mutual caring [48]. On a similar note, existing research showed that a lack of (visual) feedback from their students due to mainly turned off cameras and passive and non-responsive students made it hard for faculty to get a feeling for their class, to know whether students could follow or not [41,44,45,48–50], that is to connect with their students and feel related. Overall, we hypothesized that faculty teaching online during the COVID-19 crisis would perceive their relatedness with students being considerably impaired compared to those teaching face-to-face before and that this effect would be exacerbated, the fewer students would be visible to faculty during a synchronous, video-based online class.

Regarding the subjective well-being of faculty, we expected that it would be thwarted during emergency online teaching, that is we expected that faculty would report substantially less positive emotions and more negative emotions as well as overall reduced teaching satisfaction. This idea is derived from the assumption that the associations between the satisfaction of the three basic psychological needs for autonomy, competence, and relatedness and the different aspects of well-being in teaching professions [16,18,28–32] also apply to online teaching, as well as from the assumption that the needs for both competence and relatedness would be impaired in online emergency teaching during the COVID-19 crisis, as compared to face-to-face teaching before the crisis.

In a nutshell, the deliberations above lend support for the idea that the teaching format (emergency online vs. face-to-face) may be associated with differences in the satisfaction of the three basic needs, which in turn may be associated with differences in subjective well-being. This implies a mediation of the effect of teaching format on subjective well-being through need satisfaction. To date, however, no study has systematically compared faculty teaching experiences during versus before the COVID-19 crisis or the effects of need satisfaction on different aspects of well-being in faculty teaching online.

## The present study

The key goal of the present study was to explore how faculty responded to the sudden shift from habitual face-to-face teaching on university sites to video-based synchronous online teaching by systematically comparing their teaching experiences to faculty who taught face-to-face before the pandemic. To this end, we used pre-pandemic faculty data from [51] and replicated their diary-design for the data collection during the first COVID-19-enforced emergency online teaching year and compared faculty experiences during and before the COVID-19 crisis using two approaches: 1) internal, retrospective comparison of general emergency online teaching experiences and own face-to-face teaching experiences before the pandemic rated by the online teaching sample only and 2) group comparisons between the emergency online teaching vs. face-to-face teaching sample with respect to in-class experiences.

Drawing on research of faculty members' experiences in teaching and deliberations about the exceptional circumstances when face-to-face teaching was ad hoc shifted to emergency online teaching in a time of pandemic, we expected faculty to indicate comparable levels of satisfaction of the need for autonomy, a reduced satisfaction of the need for competence, and a clearly reduced satisfaction of the need for relatedness in emergency online teaching compared to face-to-face teaching. Based on SDT [15] and an empirical foundation on links between basic need satisfaction and emotional experiences in the teaching context [18,20,22,25,29,30,32,52], we furthermore expected impaired subjective well-being, that is lower levels of the positive emotions enjoyment and pride, higher levels of the negative emotions boredom, anger, anxiety, and shame, and lower levels of teaching satisfaction in emergency online compared to face-to-face teaching. In further analyses we tested whether the satisfaction of the needs for autonomy, competence, and relatedness mediated the effect of teaching format, that is teaching synchronous online classes during the crisis versus face-to-face classes before the crisis, on subjective well-being.

Above and beyond these main effects of emergency online versus pre-crisis face-to-face teaching, we sought to take a closer look at the factors within synchronous online teaching settings that may influence need satisfaction. Specifically, we expected that more technical problems would predict lower satisfaction of the need for competence and that online environments allowing more for quasi-live video-based interaction, as quantified by the average number of participants sharing their videos during a session, would predict higher satisfaction of the need for relatedness.

## Method

This study has been pre-registered before the start of data collection of the online teaching sample during the COVID-19 pandemic. The pre-registration, data, and analysis scripts are available through https://osf.io/b9y4a/.

The research reported herein was conducted in accordance with the APA ethical standards and has received a formal waiver of ethical approval by the ethics committee of the Department of Psychology of the University of Munich. Participation in the study was voluntary, all participants gave written informed consent, and no identifiers that could link individual participants to their results were obtained. Hence, all the analyses were conducted on anonymous data.

### Procedure and measures

For Sample 1, we obtained data collected in the context of a different study ([51]; PsyArXiv: p4nhu) from faculty teaching face-to-face before the pandemic and replicated the survey design to collect corresponding data from faculty teaching online during the pandemic for

Sample 2. Sample 1 was recruited from two German universities and data was collected before the start of the study (basic questionnaire) and directly after having taught multiple classes (session-specific questionnaire). Sample 2 was recruited by sending out e-mails to the study deans of eight large German universities as well as colleagues asking them to participate in and forward our online survey invitation to their colleagues.

Faculty of both samples were asked to complete a basic questionnaire once and a session-specific questionnaire several times. All items and scales used in the basic and session-specific questionnaires were German translations or adaptations of established English-speaking scales. First, faculty answered the basic questionnaire that covered basic need satisfaction (German adaptation of [53]) and self-efficacy (German adaptation of [54]) before the pandemic (faculty of Sample 2 rated the aspects retrospectively while the pandemic had already set in), as well as faculty members' current stress at work [55]. Sample 2 was additionally asked to generally report basic need satisfaction and teaching satisfaction with respect to emergency online teaching in the current time of pandemic as judged against their own face-to-face teaching experiences before the pandemic (adapted from the session-specific questionnaire items). Subsequently, all faculty were asked to fill in a session-specific questionnaire ideally three to six times directly after having taught the same class, whereby online classes had to be taught synchronously using an online meeting tool. Sample 1 had done so for multiple on-site classes they taught that semester, but for the purpose of this study the first class of each faculty in the dataset was used for further analyses. Sample 2 was asked to do so for exactly one of the classes they currently taught, which they could choose freely. The session-specific questionnaire tapped at basic need satisfaction (adaptation of [56]), discrete emotions (based on [57]), and teaching satisfaction (self-developed by [51]). Sample 2 additionally indicated technical aspects of their online environment, including the number and approximate time fraction of activated student cameras during the session. This information was used to calculate the average number of visible students across a session. Although some more variables were collected especially from Sample 1, the current study focused on the reported measures and therefore omitted the other constructs. Measurement properties and example items of all central study variables are depicted in Table 1.

## Sample

Initially, in Sample 1 $n$ = 95 participants answered the basic questionnaire and $n$ = 101 participants answered the sessions specific questionnaire ($n$ = 89 answered both). In Sample 2, $n$ = 123 participants answered the basic and $n$ = 71 participants answered the session-specific questionnaire ($n$ = 60 answered both). For the purpose of this study, participants who had answered the basic questionnaire only but did not move on to the session-specific questionnaire were excluded from further analyses, which resulted in $N$ = 172 participants in total.

Finally, Sample 1 comprised $n$ = 101 faculty who taught face-to-face in classrooms before the pandemic (52.8% female; aged $M$ = 40.0, $SD$ = 10.4; work experience of $M$ = 9.3, $SD$ = 7.8 years; obtained from [51]). Sample 2 comprised $n$ = 71 faculty who taught online during one of the first academic terms within the first year of the COVID-19 pandemic (i.e., between March 2020 and March 2021) by offering synchronous classes using an online meeting tool, such as AdobeConnect, BigBlueButton, WebEx, or Zoom (63.3% female; aged $M$ = 39.6, $SD$ = 11.0; work experience of $M$ = 9.0, $SD$ = 8.3 years). This is the form of online teaching that resembles face-to-face teaching most closely. We explicitly did not include flipped classroom settings or asynchronous offers implemented through online learning platforms because they are too different from face-to-face teaching to make meaningful comparisons.

**Table 1. Number of items, sample items, and reliability indicators for both samples of all study variables.**

| | No. | Item stem / sample item | Cronbach's α (Donald's Ω) in Sample | |
|---|---|---|---|---|
| | | | 1 | 2 |
| Basic questionnaire | | | | |
| Experiences before the pandemic (rated retrospectively by Sample 2) | | | | |
| Basic need satisfaction | | Typically, in my teaching . . . | | |
| Autonomy [a] | 6 | I am free to do things my way. | .67 (.68) | .76 (.78) |
| Competence [a] | 6 | I also master difficult things well. | .73 (.71) | .77 (.74) |
| Relatedness [a] | 6 | I feel close and connected to colleagues who are important to me. | .78 (.85) | .76 (.84) |
| Self-efficacy [a] | 9 | Typically, in your teaching, how well do you accomplish to . . . use varied teaching methods? | .83 (.80) | .82 (.70) |
| Experiences during time of data collection | | | | |
| Stress at work [b] | 8 | How often did you experience times when you had too many commitments to fulfill? | .94 (.94) | .94 (.95) |
| Technical problems [a] | 4 | There are technical problems all the time. | – | .84 (.95) |
| General emergency online teaching experiences judged against prior experiences (Sample 2 only) | | | | |
| Basic need satisfaction | | Compared to my typical experiences in non-online teaching so far, I feel like . . . | | |
| Autonomy [c] | 2 | I can determine how I design my teaching. | – | .83 |
| Competence [c] | 2 | I can handle my teaching well and competently. | – | .86 |
| Relatedness [c] | 2 | I feel like I'm socially connected. | – | .60 |
| Teaching satisfaction [c] | 1 | I'm satisfied with my teaching. | – | – |
| Session-specific questionnaire | | | | |
| Emotions | | In today's session, I experienced. . . | | |
| Enjoyment [a] | 1 | enjoyment | – | – |
| Pride [a] | 1 | pride | – | – |
| Boredom [a] | 1 | boredom | – | – |
| Anger [a] | 1 | anger | – | – |
| Anxiety [a] | 1 | anxiety | – | – |
| Shame [a] | 1 | shame | – | – |
| Teaching satisfaction [a] | 1 | Overall, I am satisfied with today's session. | – | – |
| Basic need satisfaction | | In today's session, I felt . . . | | |
| Autonomy [a] | 2 | able to act autonomously. | .96 | .88 |
| Competence [a] | 2 | like I was competent. | .94 | .68 |
| Relatedness [a] | 2 | close and connected to my students. | .91 | .89 |

No. = Number of items.

[a] 8-point agreement scale (1 = *no agreement*, 8 = *full agreement*).

[b] 5-point rating scale (1 = *never*, 5 = *very often*).

[c] 9-point semantic differential (−4 = *less* (i.e., worse during the time of pandemic), 0 = *equal*, 4 = *more* (i.e., worse before the pandemic).

Multivariate outlier analyses revealed no multivariate outliers. The detailed data structure showing which questionnaire had been answered how often is depicted in S1 Table.

## Statistical analyses

For the analyses in the present study, the manifest values from the session-specific questionnaires were aggregated across all available sessions per participant. Data was analyzed with R [58], using Welch's independent and one-sample *t*-tests and simple linear regressions. To evaluate the significance of our findings, we focused on effect sizes rather than the significance level of $\alpha < 0.05$ and complemented the frequentist approach with the determination of Bayes factors (BF). We considered effect sizes of Cohen's *d* above *d* = .2 as small, above *d* = .5 as

medium, and above $d = .8$ as large effects [59]. A Bayes factor indicates the likelihood of the alternative hypothesis compared to the null hypothesis given the observed data, that is, a BF of 5 would indicate that the alternative hypothesis is five times more likely than the null hypothesis given the data. To interpret the results, the following rules were applied: a BF of 1–3 was considered as anecdotal or weak evidence, a BF of 3–30 as positive to strong evidence, a BF of 30–150 as strong to very strong evidence, and a BF of $>$ 150 as decisive evidence [60]. Mediation analyses were performed using the PROCESS macro [61], which estimates direct effects (effect of teaching format on criterion variable controlling for the mediator variables), specific indirect effects (effect of teaching format on criterion variable through one specific mediator variable), and total indirect effects (mediation of the effect of teaching format on criterion variable by all mediators) using a path-analytic framework. We used 10'000 bootstrap samples for the computations and considered indirect effects as significant when a bootstrap confidence interval did not include 0.

## Results

Results of all mean level comparisons are depicted in Table 2. There were no statistically significant differences between the samples with respect to gender, age, work experience, and working conditions before the pandemic, such as satisfaction of the basic needs for autonomy, competence, and relatedness, as well as self-efficacy. Faculty teaching before and during the pandemic experienced comparable levels of stress at work, had a comparable number of weekly teaching hours and spent a comparable amount of time on teaching, while faculty in Sample 2 spent slightly less time on research (small effect size, anecdotal evidence as judged by the BF).

When contrasting their emergency online teaching experiences during the time of pandemic against their own face-to-face teaching experiences before the pandemic, faculty of Sample 2 reported to experience comparable levels of autonomy and competence, but clearly reduced levels of relatedness (large effect size, decisive evidence as judged by the BF), and slightly reduced levels of teaching satisfaction (small to medium effect size, anecdotal evidence as judged by the BF).

The sample comparisons showed that faculty teaching online during a time of pandemic reported to experience less autonomy (medium effect size, strong evidence as judged by the BF), less competence and teaching satisfaction (medium effect sizes, positive evidence as judged by the BF), and clearly less relatedness (medium effect size, decisive evidence as judged by the BF) compared to those teaching before the pandemic. Furthermore, faculty teaching online reported to experience substantially less enjoyment (medium to large effect size, decisive evidence as judged by the BF), more anger (medium effect size, strong evidence as judged by the BF), slightly more shame (small to medium effect size, anecdotal evidence as judged by the BF), and comparable levels of pride, boredom, and anxiety (small effect sizes, evidence in favor of null hypothesis as judged by the BFs), compared to faculty teaching face-to-face before the pandemic. Fig 1 depicts the central results from the within- and between participant comparisons.

Regression analyses within the emergency online teaching sample indicated that technical problems did not predict perceived competence ($β = −.15$, $R^2 = .02$, $p = .24$, BF = 0.47, $n = 60$) and that the number of students that were visible on average during a synchronous online class only tended to influence faculty members' perceived relatedness with students ($β = .26$, $R^2 = .07$, $p = .03$, BF = 2.04, $n = 71$).

Overall, bivariate correlations among the basic needs, discrete teaching emotions, and teaching satisfaction as measured directly after teaching a class were in line with SDT, that is

**Table 2. Mean level comparisons of all study variables.**

| | Sample 1 | | Sample 2 | | | | | |
|---|---|---|---|---|---|---|---|---|
| | *M* | *SD* | *M* | *SD* | *t* | *p* | *d* | BF |
| Sample characteristics | | | | | | | | |
| Age | 40.01 | 10.42 | 39.57 | 10.95 | −0.25 | .806 | 0.04 | 0.19 |
| Work experience | 9.33 | 7.84 | 9.01 | 8.28 | −0.23 | .815 | 0.04 | 0.18 |
| Stress at work | 3.18 | 0.94 | 3.24 | 0.95 | 0.36 | .719 | 0.06 | 0.19 |
| Weekly teaching hours | 6.75 | 4.21 | 6.76 | 4.64 | 0.01 | .991 | 0.00 | 0.18 |
| Time spent on teaching | 17.15 | 10.29 | 19.42 | 13.93 | 1.08 | .284 | 0.19 | 0.33 |
| Time spent on research | 19.31 | 12.10 | 14.32 | 12.04 | −2.47 | **.015** | 0.41 | 2.81 |
| Working conditions before the pandemic[a] | | | | | | | | |
| Autonomy[a] | 5.97 | 1.02 | 6.01 | 0.97 | 0.24 | .809 | 0.04 | 0.18 |
| Competence[a] | 6.20 | 0.97 | 6.27 | 0.87 | 0.49 | .624 | 0.08 | 0.20 |
| Relatedness[a] | 6.09 | 1.32 | 6.41 | 1.15 | 1.54 | .126 | 0.25 | 0.50 |
| Self-efficacy[a] | 5.94 | 0.89 | 5.99 | 0.80 | 0.38 | .708 | 0.06 | 0.19 |
| Emergency online teaching experiences judged against prior experiences (Sample 2) | | | | | | | | |
| Autonomy | – | – | 0.33 | 1.62 | 1.60 | .116 | 0.21 | 0.47 |
| Competence | – | – | 0.05 | 1.38 | 0.28 | .780 | 0.04 | 0.15 |
| Relatedness | – | – | −1.97 | 1.46 | −10.41 | **< .001** | 1.34 | 1.2e[a] |
| Teaching satisfaction | – | – | −0.55 | 1.74 | −2.45 | **.017** | 0.32 | 2.18 |
| Session-specific teaching experiences | | | | | | | | |
| Enjoyment | 6.51 | 0.99 | 5.70 | 1.45 | −4.10 | **< .001** | 0.68 | 823.03 |
| Pride | 4.01 | 1.54 | 4.07 | 1.79 | 0.25 | .803 | 0.04 | 0.17 |
| Boredom | 2.06 | 1.02 | 2.37 | 1.36 | 1.61 | .109 | 0.26 | 0.63 |
| Anxiety | 1.55 | 0.82 | 1.83 | 1.37 | 1.50 | .137 | 0.25 | 0.57 |
| Anger | 1.65 | 0.78 | 2.32 | 1.62 | 3.23 | **.002** | 0.56 | 59.85 |
| Shame | 1.37 | 0.63 | 1.75 | 1.29 | 2.26 | **.026** | 0.39 | 3.05 |
| Teaching satisfaction | 6.55 | 0.90 | 6.01 | 1.29 | −3.06 | **.003** | 0.50 | 20.21 |
| Autonomy | 7.09 | 0.96 | 6.51 | 1.15 | −3.48 | **< .001** | 0.56 | 56.10 |
| Competence | 6.81 | 0.93 | 6.32 | 1.02 | −3.23 | **.002** | 0.51 | 22.40 |
| Relatedness | 5.44 | 1.30 | 4.55 | 1.67 | −3.78 | **< .001** | 0.61 | 184.98 |

Negative *t*-values indicate lower values of the respective variables in emergency online teaching during the pandemic than in face-to-face teaching before the pandemic.
[a] Rated retrospectively by Sample 2.

need satisfaction positively correlated with positive emotions and teaching satisfaction correlated negatively with negative emotions (see S2 Table). Mediation analyses supported our expectations that the effects of the teaching format (online vs. face-to-face) on subjective well-being were mediated by the satisfaction of the basic needs, as indicated by significant total indirect effects of the teaching format through the satisfaction of the three basic needs on all outcome variables except anxiety (for detailed results see Fig 2). Specifically, the effect of the teaching format on enjoyment was mediated by autonomy and relatedness, the effect on pride was mediated by relatedness, the effects on anxiety and satisfaction were mediated by competence and relatedness, the effect on anger was mediated by autonomy and relatedness, and the effect on shame was mediated by competence, in SDT-congruent ways.

## Discussion

The present study compared the satisfaction of the three basic psychological needs for autonomy, competence, and relatedness as well as subjective well-being of faculty teaching online

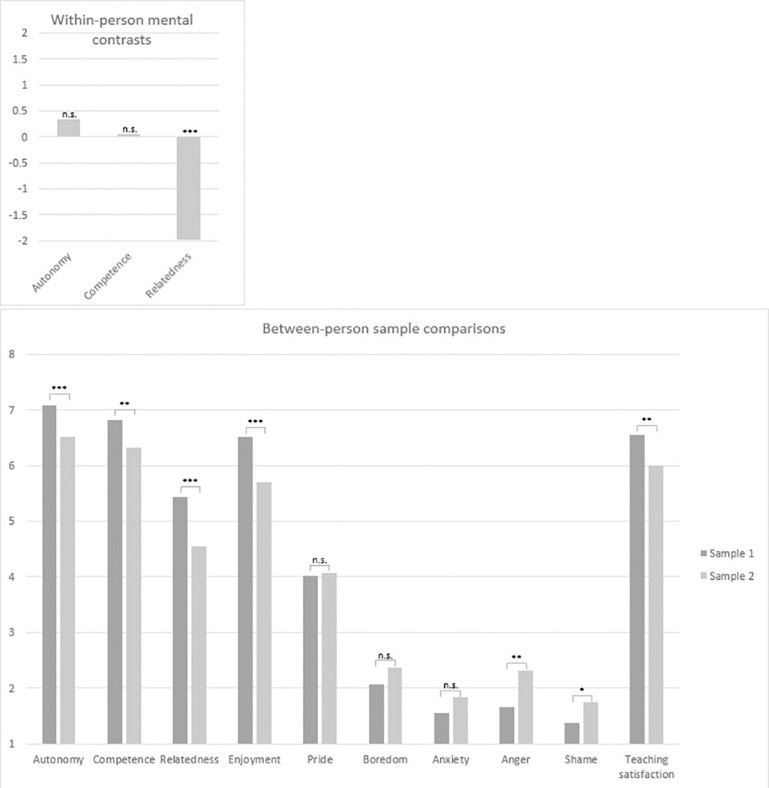

**Fig 1. Face-to-face versus online teaching experiences: Mean level comparisons of the within-person mental contrasts and the between-person sample comparisons.** The within-person mental contrasts refer to emergency online teaching experiences judged against own prior experiences (Sample 2 only), whereby 0 indicates comparable, positive values indicate better, and negative values indicate worse need satisfaction during the crisis, respectively. Between-person sample comparisons refer to experiences of Sample 1 (teaching face-to-face before the crisis) compared to experiences of Sample 2 (teaching online during the crisis). *** $p < .001$. ** $p < .01$. * $p < .05$.

after the onset of the COVID-19 crisis in spring 2020 to faculty teaching face-to-face before the crisis.

Overall, both samples were well comparable, the only small but interesting difference regarded the hours spent on research during the time of data collection, which was less during the time of pandemic. As the hours spent on teaching were comparable, this may imply that during the pandemic either research efforts were impaired directly, for instance because testing in laboratories was impossible due to contact restrictions and hygiene regulations, or that less time could be spent on research due to other responsibilities such as child- and elderly-care, or increased efforts for administrative and organizational tasks.

The central findings of the present study are that faculty clearly suffered from decreased relatedness with their students and reduced teaching enjoyment when the COVID-19 crisis hit and they had to switch to purely online teaching. Furthermore, SDT is a promising framework to approach faculty well-being, including discrete emotions and teaching satisfaction, in online teaching.

## Reduced basic need satisfaction in emergency online versus face-to-face teaching

Surprisingly, faculty teaching online during the pandemic compared to faculty teaching before the pandemic reported impaired satisfaction not only of the needs for competence and

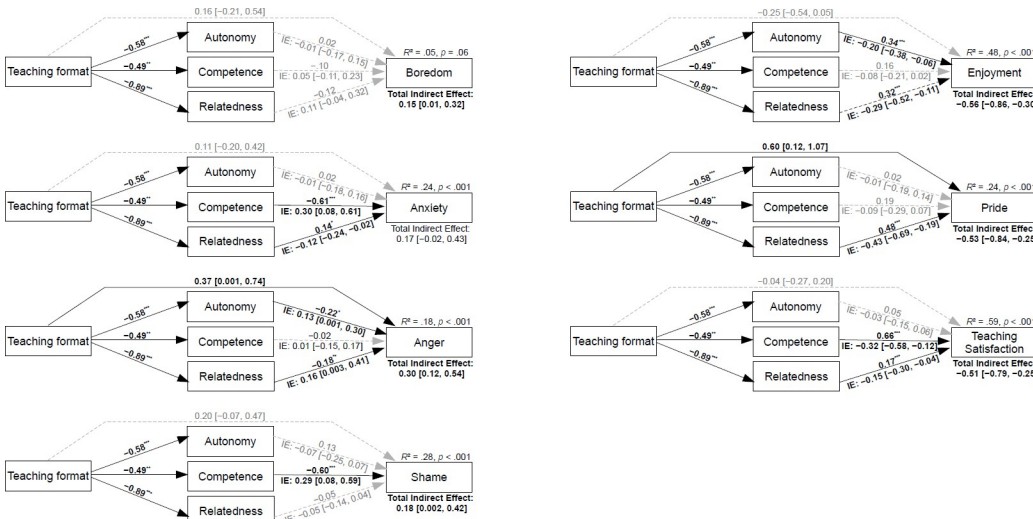

**Fig 2. Mediation analyses results.** Mediation analyses depict unstandardized coefficients, which indicate differences in the mediator variables that are due to the teaching format (emergency online vs. face-to-face) and differences in the criterion variable that are due to differences in the mediator variable. Negative values from teaching format to need satisfaction indicate lower values in the emergency online teaching sample. IE = Indirect effect of teaching mode through the basic need on the criterion variable incl. bootstrap confidence interval; for a significant indirect effect, the respective CI does not include 0. Solid black lines depict significant, dashed grey lines non-significant effects. *** $p < .001$. ** $p < .01$. * $p < .05$.

relatedness but also for autonomy. The reduction in relatedness between faculty teaching before and during the crisis was also supported by means of mental contrasting between in-crisis and pre-crisis teaching within the COVID-19-sample, but the reduction in perceived autonomy and competence were not.

We propose that this differential pattern of findings for competence and autonomy can be explained by the different measurement approaches. While the between-person sample comparison with other faculty teaching face-to-face before the pandemic compared in-situ experiences within teaching sessions, the within-person mental contrast involved faculty members' more general beliefs about their own teaching experiences (see also [62], on discrepancies across state vs. trait self-reports of teaching emotions). For instance, with respect to perceived autonomy when teaching in general, faculty may have focused on their autonomy in choosing content, their preferred implementation of online teaching, or their gained flexibility [39,46,50] with respect to time and location, rather than on session-specific limitations, such as fewer teaching methods, when contrasting their current against previous experiences. With respect to competence, in making such a general comparison faculty may have focused on stable aspects of their own competence that apply to both online and face-to-face teaching, such as content knowledge. When judging their in-situ experiences right after having delivered a class online during the pandemic, however, technical hassles and restrictions implied by the digital format may have been more salient and may therefore have undermined their in-situ experiences of competence. Although technical problems did not significantly predict the satisfaction of the need for competence in our study, very likely due to measurement-inherent problems because technical problems were assessed generally and not situation-specific, they still seem to be a promising factor in promoting or thwarting online teaching competence when assessed situation-specific, which also aligns with notions on faculty readiness to teach online [43,52].

The varying results depending on the measurement approach, that is judging against own experiences versus comparing two groups, show that it does probably not describe the whole picture when research solely relies on general retrospective judgements that are supposed to compare experiences during the highly exceptional situation of the COVID-19 crisis to experiences before the crisis and that such findings need to be interpreted with caution.

The fact that the physical distance between faculty and their students was so salient that faculty consistently experienced severely reduced levels of perceived relatedness, irrespective of the measurement approach, speaks to the robustness of this finding, which is further corroborated by similar findings from studies conducted during the COVID-19 crisis without a control group (e.g., [48,50]). We consider this one of the key findings of the present study and propose that before the COVID-19 pandemic, the importance of relatedness in the higher educational context may have been underestimated, because it develops rather easily in face-to-face settings when regularly interacting with and getting to know each other [63,64]. Although there are possibilities to form relationships with students in online contexts, for instance by self-disclosure, that is revealing personal information, responding in a timely manner, and using humor [65,66], such offers probably cannot fully compensate the loss of recurrent classroom interactions that more naturally allow to develop mutual relationships between individual students and faculty members [63,64]. Our data could only provide weak evidence that the number of visible students during a class may be systematically linked with relatedness, which was probably due to a highly limited range in the number of visible students. Nevertheless, it should still be considered as possible factor that contributes to creating a feeling of relatedness in or preference for online teaching settings [39].

To sum up, while faculty were very well aware that their need for relatedness was thwarted in emergency online as compared to face-to-face teaching, they did not perceive their own teaching autonomy and competence being reduced by the shift to emergency online teaching, although the in-class comparisons hint in the direction that teaching autonomy and competence were indeed reduced compared to before the crisis. Subjective well-being in emergency online versus face-to-face teaching and implications for theory

In line with expectations, faculty teaching online during the pandemic reported impaired subjective well-being, that is they reported to experience considerably less enjoyment and teaching satisfaction but increased anger and with a tendency also more shame than faculty teaching face-to-face before the pandemic. This confirms our expectation that the COVID-19 crisis considerably impaired not only students' (e.g., [67–70]), but also faculty member's emotional experiences and satisfaction as implied by the sudden shift to exclusively teaching online. Further in line with expectations, the effects of the teaching format on the different aspects of subjective well-being were overall mediated by the satisfaction of the basic needs for autonomy, competence, and relatedness in ways that are congruent with propositions from SDT [12]. That is, the reduced levels of positive emotions and teaching satisfaction as well as the increased levels of negative emotions in online compared to face-to-face teaching could mostly be explained by reduced relatedness and typically either reduced autonomy or competence in online compared to face-to-face teaching.

It is worth noting, though, that certain discrete emotions, namely pride, anxiety, and boredom, were comparable in emergency online teaching and face-to-face teaching before the crisis. We speculate that this is because discrete emotions are triggered by a variety of appraisal processes, as suggested for instance by the control-value theory of achievement emotions [71], which are not included in SDT. Control-value theory proposes that differential combinations of control and value appraisals trigger specific discrete emotions, for example high control and high value are supposedly associated with enjoyment and pride, and low control and high

value are supposedly associated with anxiety (see [71]). This line of reasoning may help in understanding the unexpected findings.

First of all, pride was not reduced in emergency online compared to face-to-face teaching. According to control-value theory, pride is claimed to be elicited when successfully mastering an activity [71]. Possibly, taking pride in mastering teaching could take different forms: faculty may have been proud when they managed to deliver high quality teaching in face-to-face settings, but they may have been just as proud when they accomplished to teach online despite the challenging circumstances. Such subjective definitions of success may explain why pride in online compared to face-to-face teaching was not reduced. Nevertheless, our findings confirmed that the degree to which teachers felt related to their students was linked to their pride, which is in line with earlier research proposing that establishing a relationship with students is one aspect of successful teaching [72], which is seemingly accomplished easier in face-to-face than in online settings.

Second, boredom was not enhanced in emergency online compared to face-to-face teaching. Boredom is claimed to arise in repetitive tasks that are not at an optimal level of challenge and lack value [71]. Because teaching is a highly diverse task that holds a lot of variety and typically requires a highly active role of faculty, teaching seems to be a task that generally rarely triggers boredom [73], likely irrespective of the teaching format in contrast to work in general or learning, which may encompass more phases of repetitive tasks and inactivity.

Lastly, there were no group differences for anxiety. This emotion has been shown to be mainly triggered by reduced competence [22], which aligns with our findings from the mediation analysis. We propose that the anxiety-promoting effects of lowered competence in the digital teaching context were compensated by the concurrent lowered relatedness, because the lower perceived relatedness in online teaching decreased anxiety. In other words, as much as faculty may have felt insecure in the online environment, they may not have cared as much as in face-to-face contexts where they literally had to "look their students in the eyes" during teaching, resulting in overall zero effects of online versus face-to-face teaching on anxiety.

Taken together, our results show that emergency online teaching in a time of pandemic was a clearly less pleasurable activity as compared to teaching face-to-face before the pandemic (also noted for instance by [10,35,69]), which can be attributed specifically to severely reduced need satisfaction, especially of the need for relatedness. Nevertheless, it needs to be considered that need satisfaction alone did not fully explain the emergence of specific discrete emotions. It seems a promising avenue to explore the relationships between self-determination theory and control-value theory in more detail, that is to connect and supplement SDT with control and value appraisals. Generally, satisfaction of the basic needs may be considered as antecedents of control and value appraisals [74]: autonomy may very well be positively linked with both control and value appraisals, competence may be positively linked with control appraisals, and relatedness may be positively linked with value appraisals (for first empirical support for such claims in a student sample, see [75]). Taken together, it would be interesting to find out whether the satisfaction of the three basic needs indeed triggers specific control and value appraisals, which in combination trigger specific discrete emotions in general and in online teaching settings in particular.

### Implications for post-pandemic online teaching

Overall, the question emerges to what degree the present results can be generalized to post-crisis online teaching. The onset of the COVID-19 pandemic was an unprecedented time that challenged faculty not only at the workplace but also in private life. Especially in the beginning of the pandemic, the workload to design classes and the job demands in general were very high

due to for example a lack of familiar online teaching approaches, lack of technical competence, and the need to adapt working habits, and institutions were partially unable to react to the new circumstances quickly enough to provide high quality support, while at the same time contact restrictions dramatically reduced opportunities for exchange with colleagues and students. Such factors have generally been related to lower levels of need satisfaction [76] and in such exceptional times they may have contributed to even more severely impaired need satisfaction than they would have in normal times. On top of these burdens at the workplace, there were omnipresent stressors outside the job context, such as child- and elderly care, home schooling, lack of contact with and worries about one's health and that of friends and family, which may have carried over to the experiences when teaching to some extent.

It thus seems reasonable, that need satisfaction would not be impaired as severely in post-crisis online teaching, because faculty likely would complement their online-courses with individual face-to-face formats that allow for the establishment of interpersonal relations. To nevertheless tackle the challenge of reduced relatedness not only between faculty and students but also among students, it is necessary to actively foster interaction and timely communication to create a feeling of belonging when teaching and learning online [66]. Some possibilities to do so may be to encourage students to share their videos during discussions in order to support the feeling of being in a class together, and to trigger self-disclosure by for instance prompting students to talk about an informal personal topic in a small group before starting content-focused groupwork or discussions in breakout sessions [50,66,77,78].

Further, in the meantime, and hopefully by the time the pandemic is fully overcome, many faculty will have gained extensive experience in and developed skills for online teaching, specifically in overcoming technical hassles and enriching the online formats with adequate interactive activities, which should result in higher teaching autonomy and competence. Nevertheless, institutional and mainly individualized support in designing new online or hybrid classes [42] as well as regular exchange about teaching experiences among faculty seem imperative to steadily improve online teaching offers.

Taken together, the predictions of SDT and the positive prospects for the development of need satisfaction in online teaching over time also lend support for the claim that the subjective well-being of faculty in post-pandemic online teaching will be better than in emergency online teaching, qualified by more positive and less negative emotional experiences, as well as higher teaching satisfaction.

## Limitations and directions for future research

One limitation of the present study is the relatively small sample size of faculty teaching online during the pandemic. Nevertheless, the presented findings are informative because of the very similar sample composition in terms of for example age, gender, and teaching experience of faculty teaching face-to-face before and those teaching online during the COVID-19 crisis.

Furthermore, the emotional experiences when teaching during the pandemic may have been colored to some extent by general emotional experiences due to the overall burdening situation especially during the first months of the COVID-19 crisis. Yet it is worth considering that teaching emotions have been shown to be highly context-specific [79–81], thus these effects were probably not very pronounced.

Within the scope of this study, it was not possible to focus on the many different specific factors that may hinder or foster the satisfaction of the basic needs and subjective well-being in online teaching. We propose that it is a promising avenue for future research to identify the origins of need satisfaction and emotions in online teaching to develop elaborated recommendations on how to make online teaching more attractive and enjoyable for faculty and

students. To this end, future research could assess various factors in synchronous and asynchronous online environments that may influence need satisfaction, such as perceived limitations in useable teaching methods, technical problems experienced during a session rather than technical problems in general as assessed in this study, perceived active participation and responsiveness of students, as well as communication forms rather than the mere quantification of number of students sharing their videos as in this study. Knowledge about factors that foster need satisfaction and positive emotional experiences may enable faculty to develop classes that blend the best of both digital and in-person teaching approaches. Such classes may comprise face-to-face meetings that are enriched by technology, such as live polling, etherpads, or online mind maps that allow for interaction even in larger lecture-size groups [41], as well as asynchronous online teaching units, thus optimizing the opportunities for different ways of personal and digitally mediated interaction, combined with self-directed and therefore flexible learning phases.

## Conclusion

Although online classes can substitute rather well for some aspects of face-to-face teaching and learning, it became very obvious during the COVID-19 pandemic that higher education institutions are not only a place of knowledge generation, transmission, and advancement, but also a place that enables people to connect with each other, build relationships, and interact as social beings. Therefore, even though online education has advantages, the crucial role of a successful social integration into the university community, which probably happens more easily in face-to-face settings, must not be underestimated. In the long run, we propose that online teaching will supplement rather than replace face-to-face teaching, and we are convinced it can become a valuable addition to increase the flexibility in teaching and learning in and the access to higher education.

## Supporting information

**S1 Table. Data structure of the answered basic and session-specific questionnaires.** Data of Sample 1 (face-to-face teaching) was obtained before and data of Sample 2 (online teaching) during the COVID-19 pandemic. Participants were instructed to answer the session-specific questionnaire 3 to 6 times.
(DOCX)

**S2 Table. Correlations among all study variables for the complete sample and faculty who filled in the session-specific questionnaire several times.** AUT = autonomy, COM = competence, REL = relatedness, SE = self-efficacy, TS = teaching satisfaction, STR = stress, TP = technical problems, VS = visible students (on average), ENJ = enjoyment, PRI = pride, BOR = boredom, ANX = anxiety, ANG = anger, SHA = shame. 1–4 are reported as experienced before (B) the pandemic. 5–8 are general emergency online teaching experiences during the pandemic compared (C) against own experiences before the pandemic (Sample 2 only). 9–10 are general (G) experiences during time of data collection. 12–23 are session-specific (S) information and experiences. * p < .05. ** p < .01. *** p < .001.
(DOCX)

## Author Contributions

**Conceptualization:** Carolin Schwab, Anne C. Frenzel, Anton K. G. Marx.

**Data curation:** Carolin Schwab.

**Formal analysis:** Carolin Schwab.

**Investigation:** Carolin Schwab.

**Methodology:** Carolin Schwab.

**Project administration:** Carolin Schwab, Anton K. G. Marx.

**Supervision:** Anne C. Frenzel.

**Writing – original draft:** Carolin Schwab.

**Writing – review & editing:** Carolin Schwab, Anne C. Frenzel, Martin Daumiller, Markus Dresel, Oliver Dickhäuser, Stefan Janke.

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
