## [Decision Letter · Decision Letter 0]

20 Apr 2022

PONE-D-22-04848“I’m tired of black boxes!”: A systematic comparison of faculty well-being and need satisfaction before and during the COVID-19 crisisPLOS ONE

Dear Dr. Schwab,

Thank you for submitting your manuscript to PLOS ONE. After careful consideration, we feel that it has merit but does not fully meet PLOS ONE’s publication criteria as it currently stands. Therefore, we invite you to submit a revised version of the manuscript that addresses the points raised during the review process.

Please revise your manuscript by addressing the reviewer comments. Please note that Reviewer 2 comments are listed in the attached MS Word document. 

We look forward to receiving your revised manuscript.

Kind regards,

Heng Luo, Ph.D.

Academic Editor

PLOS ONE

Journal Requirements:

2. In the ethics statement in the Methods and online submission information, please ensure that you have specified (1) whether consent was informed and (2) what type you obtained (for instance, written or verbal, and if verbal, how it was documented and witnessed). If your study included minors, state whether you obtained consent from parents or guardians. If the need for consent was waived by the ethics committee, please include this information.

Reviewers' comments:

Reviewer's Responses to Questions

**Comments to the Author**

1. Is the manuscript technically sound, and do the data support the conclusions?

Reviewer #1: Partly

Reviewer #2: Yes

2. Has the statistical analysis been performed appropriately and rigorously? 

Reviewer #1: Yes

Reviewer #2: Yes

3. Have the authors made all data underlying the findings in their manuscript fully available?

Reviewer #1: Yes

Reviewer #2: Yes

4. Is the manuscript presented in an intelligible fashion and written in standard English?

Reviewer #1: Yes

Reviewer #2: Yes

5. Review Comments to the Author

Reviewer #1: The present study systematically compared the subjective well-being of faculty teaching face-to-face before to those teaching online during the COVID-19 pandemic, and determined the mediation effects of basic need fulfillment in teaching format on subjective well-being. Results showed some interesting differences. However, there are several deficiencies in this study.

First, the present study compares the differences between sample 1 and sample 2 about the working conditions of face-to-face teaching and teaching online during the COVID-19 pandemic. This means that it is better to repeat the measurements on the same sample. But this study reported that data collected in the context of a different study before the pandemic (Sample 1, teaching face-to-face) and replicated the survey design to collect corresponding data during the pandemic (Sample 2, teaching online). Sample 2 asked to generally report basic need fulfillment and teaching satisfaction with respect to emergency online teaching in the current time of pandemic, additionally asked to report retrospectively mentioned above during teaching face-to-face before the pandemic. Sample 1 and sample 2 appear to be different groups.

Second, for the measurement items of variables, especially basic need fulfillment, self-efficacy, emotion and teaching satisfaction, it is recommended to make it clear whether the report is adapted from a recognized scale or self-made.

Third, are 101 (sample 1) and 71 (sample 2) qualified sample sizes? If not, the authors are recommended to report the total number of participants of the two samples. At the same time, it should be explained how the researchers collected the questionnaire, such as participants selection, operation of questionnaire distribution, informed consent or ethical protection.

Fourth, the chapters of “SDT and faculty well-being” and “Faculty experiences in response to the COVID-19 crisis onset” can be integrated as “Literature Review”, which will be more normative and readable. Besides, in the chapter of Literature Review, the relationships among basic need fulfillment, teaching format and subjective well-being should be further interpreted.

Fifth，considering improving the chapter of “Introduction” to further illustrate the value of this study to higher educators’ online teaching or blended teaching in the post COVID-19 pandemic era.

Sixth, the authors are recommended to establish mediation relationship based on a certain theoretical or academic viewpoints. Besides, teaching format as a construct, including kinds of aspects. It is suggested to quantify the "teaching model" into more specific indicators for mediating effect analysis.

Reviewer #2: The authors are suggested to make some improvements before publication (More review comments are attached).

For instance, the authors should streamline the Introduction. To improve readability, the authors may benefit from focusing each paragraph on a single topic that supports the study rationale and ending with a sentence telling the reader what is to be concluded from that paragraph.

6. PLOS authors have the option to publish the peer review history of their article (what does this mean?). If published, this will include your full peer review and any attached files.

Reviewer #1: No

Reviewer #2: No

---

## [Author Response · Author response to Decision Letter 0]

7 Jun 2022

Dear reviewers,

thank you very much for your very important and constructive comments.

We hope that we could address all aspects adequately.

Please find our repsonses to your raised points below and in the separate file.

Thank you for your time and effort!

REVIEWER #1

The present study systematically compared the subjective well-being of faculty teaching face-to-face before to those teaching online during the COVID-19 pandemic, and determined the mediation effects of basic need fulfillment in teaching format on subjective well-being. Results showed some interesting differences. However, there are several deficiencies in this study.

Author Response: Thank you for your generally positive evaluation of our study and for raising a number of constructive comments which helped us to revise the paper. Below, we outline how we responded to each of your comments.

First, the present study compares the differences between sample 1 and sample 2 about the working conditions of face-to-face teaching and teaching online during the COVID-19 pandemic. This means that it is better to repeat the measurements on the same sample. But this study reported that data collected in the context of a different study before the pandemic (Sample 1, teaching face-to-face) and replicated the survey design to collect corresponding data during the pandemic (Sample 2, teaching online). Sample 2 asked to generally report basic need fulfillment and teaching satisfaction with respect to emergency online teaching in the current time of pandemic, additionally asked to report retrospectively mentioned above during teaching face-to-face before the pandemic. Sample 1 and sample 2 appear to be different groups.

Author Response: We agree that a longitudinal design would have been meaningful to detect within-person changes as a result of the onset of the pandemic. Yet with an event as unpredictable as a worldwide pandemic, it is virtually impossible to deliberately design such a study. Furthermore, due to data protection reasons, no personal records of the participants from Study 1 were available after the onset of the pandemic which is why we could not reach out to them to ask them for a repeated participation in our Study 2. Therefore, we decided to recruit a new sample, while making sure to assess a range of variables characterizing the sample (including age, work experience in years, weekly teaching hours) to assure that both samples were sufficiently comparable and any differences across them could be attributed to the emergency online teaching conditions due to the pandemic rather than any systematic sampling effects.

From your comment, though, we realize that we might not have been fully clear about the sampling design and procedure in the original version of our manuscript. We indeed looked at two different samples, one teaching face-to-face classes before the onset of the pandemic (Sample 1) and one teaching online classes during the pandemic (Sample 2). We were interested in finding out whether faculty experiences differed between face-to-face and online teaching. We chose two approaches to compare face-to-face vs. online teaching: 

1) internal, retrospective comparison of general emergency online teaching experiences against own face-to-face teaching experiences before the pandemic (Sample 2 only) and 2) group comparison (Sample 1 vs. Sample 2) with respect to in-class experiences. 

In response to your comment and to clarify this issue, we revised our text so that this information on study design and procedure has been included more explicitly in the “The present study” Section. Furthermore, we added a Figure to visually better depict the two approaches (Fig 1)

The revised text reads as follows: 

The key goal of the present study was to explore how faculty responded to the sudden shift from habitual face-to-face teaching on university sites to video-based synchronous online teaching by systematically comparing their teaching experiences to faculty who taught face-to-face before the pandemic. To this end, we used pre-pandemic faculty data from [51] and replicated their diary-design for the data collection during the first COVID-19-enforced emergency online teaching year and compared faculty experiences during and before the COVID-19 crisis using two approaches: 1) internal, retrospective comparison of general emergency online teaching experiences and own face-to-face teaching experiences before the pandemic rated by the online teaching sample only and 2) group comparisons between the emergency online teaching vs. face-to-face teaching sample with respect to in-class experiences.

Second, for the measurement items of variables, especially basic need fulfillment, self-efficacy, emotion and teaching satisfaction, it is recommended to make it clear whether the report is adapted from a recognized scale or self-made.

Author Response: We agree that it is essential information to sufficiently describe the sources of the used scales. We recognize that this information has not been clear in our first version of the manuscript. 

The revised text reads as follows:

Faculty of both samples were asked to complete a basic questionnaire once and a session-specific questionnaire several times. All items and scales used in the basic and session-specific questionnaires were German translations or adaptations of established English-speaking scales. First, faculty answered the basic questionnaire that covered basic need satisfaction (German adaptation of [53]) and self-efficacy (German adaptation of [54]) before the pandemic (faculty of Sample 2 rated the aspects retrospectively while the pandemic had already set in), as well as faculty members’ current stress at work [55]. Sample 2 was additionally asked to generally report basic need satisfaction and teaching satisfaction with respect to emergency online teaching in the current time of pandemic as judged against their own face-to-face teaching experiences before the pandemic (adapted from the session-specific questionnaire items). Subsequently, all faculty were asked to fill in a session-specific questionnaire ideally three to six times directly after having taught the same class, whereby online classes had to be taught synchronously using an online meeting tool. Sample 1 had done so for multiple on-site classes they taught that semester, but for the purpose of this study the first class of each faculty in the dataset was used for further analyses. Sample 2 was asked to do so for exactly one of the classes they currently taught, which they could choose freely. The session-specific questionnaire tapped at basic need satisfaction (adaptation of [56]), discrete emotions (based on [44]), and teaching satisfaction (self-developed by [51]). Sample 2 additionally indicated technical aspects of their online environment, including the number and approximate time fraction of activated student cameras during the session. This information was used to calculate the average number of visible students across a session. Although some more variables were collected especially from Sample 1, the current study focused on the reported measures and therefore omitted the other constructs. Measurement properties and example items of all central study variables are depicted in Table 1.

Third, are 101 (sample 1) and 71 (sample 2) qualified sample sizes? If not, the authors are recommended to report the total number of participants of the two samples. At the same time, it should be explained how the researchers collected the questionnaire, such as participants selection, operation of questionnaire distribution, informed consent or ethical protection.

Author Response: Thank you for pointing this out. We provided more detailed information on the sample (selection) and data collection procedure. 

The following information has been added in the method section: 

1) Participants

The revised text reads as follows: 

Initially, in Sample 1 n = 95 participants answered the basic questionnaire and n = 101 participants answered the sessions specific questionnaire (n = 89 answered both). In Sample 2, n = 123 participants answered the basic and n = 71 participants answered the session-specific questionnaire (n = 60 answered both). For the purpose of this study, participants who had answered the basic questionnaire only but did not move on to the session-specific questionnaire were excluded from further analyses, which resulted in N = 172 participants in total.

2) Information on ethical approval and informed consent

The revised text reads as follows: 

The research reported herein was conducted in accordance with the APA ethical standards and has received a formal waiver of ethical approval by the ethics committee of the Department of Psychology, BLINDED. Participation in the study was voluntary, all participants gave written informed consent, and no identifiers that could link individual participants to their results were obtained. Hence, all the analyses were conducted on anonymous data.

3) Information on the survey distribution

The revised text reads as follows:

For Sample 1, we obtained data collected in the context of a different study ([51]; PsyArXiv: BLINDED) from faculty teaching face-to-face before the pandemic and replicated the survey design to collect corresponding data from faculty teaching online during the pandemic for Sample 2. Sample 1 was recruited from two German universities and data was collected before the start of the study (basic questionnaire) and directly after having taught multiple classes (session-specific questionnaire). Sample 2 was recruited by sending out e-mails to the study deans of eight large German universities as well as colleagues asking them to participate in and forward our online survey invitation to their colleagues.

Fourth, the chapters of “SDT and faculty well-being” and “Faculty experiences in response to the COVID-19 crisis onset” can be integrated as “Literature Review”, which will be more normative and readable. Besides, in the chapter of Literature Review, the relationships among basic need fulfillment, teaching format and subjective well-being should be further interpreted.

Author Response: Thank you for the suggestions. We deemed it more meaningful to retain the more telling headings to structure the Literature Review rather than subsuming them under “Literature Review”, but realized that the headings were not precise enough. The new headings are now “SDT and faculty well-being in face-to-face teaching” and “Faculty need satisfaction in response to the COVID-19 crisis onset and associations with well-being in emergency online teaching”. We gladly took up your recommendations to include more explicit information on the relationships between teaching format, basic need satisfaction, and well-being.

The added text reads as follows:

Regarding the subjective well-being of faculty, we expected that it would be thwarted during emergency online teaching, that is we expected that faculty would report substantially less positive emotions and more negative emotions as well as overall reduced teaching satisfaction. This idea is derived from the assumption that the associations between the satisfaction of the three basic psychological needs for autonomy, competence, and relatedness and the different aspects of well-being in teaching professions [16,18,28–32] also apply to online teaching, as well as from the assumption that the needs for both competence and relatedness would be impaired in online emergency teaching during the COVID-19 crisis, as compared to face-to-face teaching before the crisis. 

In a nutshell, the deliberations above lend support for the idea that the teaching format (emergency online vs. face-to-face) may be associated with differences in the satisfaction of the three basic needs, which in turn may be associated with differences in subjective well-being. This implies a mediation of the effect of teaching format on subjective well-being through need satisfaction. To date, however, no study has systematically compared faculty teaching experiences during versus before the COVID-19 crisis or the effects of need satisfaction on different aspects of well-being in faculty teaching online.

Fifth，considering improving the chapter of “Introduction” to further illustrate the value of this study to higher educators’ online teaching or blended teaching in the post COVID-19 pandemic era.

Author Response: Thank you very much for pointing this out. In response, we added important new aspects to the “Introduction” Section.

The inserted text reads as follows:

Thereby, this study contributes to understanding how a worldwide stressor, such as a pandemic, affects teaching in higher education and provides information on the experiences of faculty during pandemic-enforced emergency online teaching. These insights may help to deal with and prepare for such stressors in the future and shed some light onto the factors and mechanisms that contribute to faculty well-being in online teaching in a post-pandemic era more generally.

Sixth, the authors are recommended to establish mediation relationship based on a certain theoretical or academic viewpoints. Besides, teaching format as a construct, including kinds of aspects. It is suggested to quantify the "teaching model" into more specific indicators for mediating effect analysis.

Author Response: Thank you for the input, we understand that our description and reasoning about the proposed mediation has not been clear enough in the first version of the manuscript. In response, we added a section describing it explicitly at the end of the section on “Faculty experiences need satisfaction in response to the COVID-19 crisis onset and associations with well-being in emergency online teaching” and reiterated it in the “The present study section”.

Here is a conceptual illustration of the mediation:

[available only in the word file] 

We prefer to not add this conceptual mediation model into our manuscript as we do not want to place too much emphasis on the mediation, because the differences between the samples are the core focus of the paper. However, the structure of the proposed mediation is depicted in the results in Figure1. We would be willing to integrate the conceptual Figure into the manuscript to illustrate the assumed mediation, if the reviewer and/or the editor recommend us to do so. 

The revised “The present study” section reads as follows : 

Drawing on research of faculty members’ experiences in teaching and deliberations about the exceptional circumstances when face-to-face teaching was ad hoc shifted to emergency online teaching in a time of pandemic, we expected faculty to indicate comparable levels of satisfaction of the need for autonomy, a reduced satisfaction of the need for competence, and a clearly reduced satisfaction of the need for relatedness in emergency online teaching compared to face-to-face teaching. Based on SDT [15] and an empirical foundation on links between basic need satisfaction and emotional experiences in the teaching context [18,20,22,25,29,30,32,52], we furthermore expected impaired subjective well-being, that is lower levels of the positive emotions enjoyment and pride, higher levels of the negative emotions boredom, anger, anxiety, and shame, and lower levels of teaching satisfaction in emergency online compared to face-to-face teaching. In further analyses we tested whether the satisfaction of the needs for autonomy, competence, and relatedness mediated the effect of teaching format, that is teaching synchronous online classes during the crisis versus face-to-face classes before the crisis, on subjective well-being.

The teaching format solely refers to teaching online vs. face-to-face and does not pertain to any further factors. We tried to make this a little clearer by reformulation the respective part in the “The present study” section. 

The revised text reads as follows:

In further analyses we tested whether the satisfaction of the needs for autonomy, competence, and relatedness mediated the effect of teaching format, that is teaching synchronous online classes during the crisis versus face-to-face classes before the crisis, on subjective well-being. 

REVIEWER #2 

The authors are suggested to make the following improvement before publication:

Thank you very much for reviewing our manuscript and providing many helpful and constructive suggestions.

The authors should streamline the Introduction. To improve readability, the authors may benefit from focusing each paragraph on a single topic that supports the study rationale and ending with a sentence telling the reader what is to be concluded from that paragraph.

Author Response: Thank you very much for pointing this out. We tried to address this and therefore restructured the paragraphs and added a wrap-up sentence to the paragraphs whenever possible.

For example:

- Taken together, need satisfaction seems to be associated with well-being in various settings.

- Taken together, the associations between need satisfaction and well-being seem to apply to teaching contexts in both schools and higher education. This research, however, is limited to face-to-face teaching — empirical studies on such explicit associations during online teaching are still lacking (as noted for instance by [12,13]).

Please move the following contents to the Introduction section:

The ultimate goal during the onset of the COVID-19 pandemic was to keep up teaching and learning higher education. In many western, well-developed countries (including Germany where the present study was conducted), video-based digital tools were 105 quickly available and allowed to offer synchronous teaching formats, which were, however, compromised by problems with bandwidth, poor audio quality, complex technological handling etc (From line 102 to line 107).

Author Response: Thank you for this suggestion. We saw some obstacles in implementing it, though. Firstly, moving these contents to the Introduction section would narrow the intentionally rather general introduction to online teaching during the pandemic into an already very specific (synchronous offers, German context) context, which would not fully cover the many highly different approaches to solving the issue worldwide. Secondly, the content is needed to transition from the application of self-determination theory in higher education in face-to-face settings to its applicability also in online settings.

To nevertheless address this suggestion, parts of the content were moved to the “Introduction” section and adapted and the remaining parts shortened.

The revised text reads as follows:

Introduction: 

The onset of the COVID-19 pandemic in early 2020 implying public lock-downs aimed at mitigating the spread of the new virus inevitably also hit universities worldwide. They had to close down lecture halls and very suddenly stop the still predominant face-to-face teaching [4,5]. In striving to keep up teaching and learning in higher education, countries worldwide tried to move to online teaching as quickly as possible and many aimed for video-based digital tools to allow the offering of synchronous online teaching formats [6–8].

Kept original section:

At the onset of the COVID-19 pandemic, in many western, well-developed countries including Germany where the present study was conducted, video-based digital tools were quickly available and allowed to offer synchronous online teaching formats. Such formats, however, can be compromised by problems such as bandwidth, poor audio quality, and complex technological handling [10,33].

Please move the following contents to the Present Study section:

The key goal of the present study was to explore how faculty responded to these sudden shifts to video-supported synchronous online teaching, as compared to the habitual face-to-face teaching on university sites, in terms of their basic need satisfaction and subjective well-being (From line 107 to line 110).

Author Response: Thank you very much for this valuable suggestion. In response, we moved the corresponding content to the beginning of the “The present study” section, with few adaptations. 

The revised text reads as follows: 

The key goal of the present study was to explore how faculty responded to the sudden shift from habitual face-to-face teaching on university sites to video-based synchronous online teaching by systematically comparing their teaching experiences to faculty who taught face-to-face before the pandemic. To this end, we used pre-pandemic faculty data …

From line 113 to line 133, almost all the contents are given by the author. To improve the persuasiveness of the arguments, the author should refer to more literatures to justify the arguments.

Author Response: Thank you for pointing this out. We included references wherever possible. Unfortunately, there is not so available yet.

The revised text reads as follows:

Regarding the satisfaction of the need for autonomy, to the degree that faculty experienced a lack of freedom in determining the content, activities, or policies in class, their perceived autonomy could be thwarted, as has been shown for graduate teaching assistants in face-to-face teaching [36]. During the onset of the COVID-19 crisis, on the one hand, the rapid shift from familiar face-to-face to unfamiliar online teaching may have impaired faculty members’ perceived autonomy because the new format may not have aligned very well with their ideas about teaching, may have made their habitually used face-to-face teaching methods inapplicable, and may have imposed the challenge to develop new teaching methods, thus possibly reducing autonomy within teaching sessions (e.g., [37]). On the other hand, due to the unprecedented circumstances, faculty were typically offered maximum flexibility when it came to maintaining their teaching activities and could decide how to offer their classes with sample options ranging from interactive, synchronous sessions using an online meeting tool to asynchronous, purely text-based self-learning units (e.g., [10]). This choice likely even increased their freedom with respect to workplace and time management because online courses do not require physical presence at university and asynchronous offers do not even require attendance at a specific time. Assuming that faculty presumably chose the online teaching approach that fit their own preferences, teaching conceptions, and competencies best (e.g., [38]) and that they gained freedom with respect to workplace and time during their work days ([39]; analogous to findings in a student sample; see [40]), their perceived autonomy should not have been impaired because the transition to emergency online teaching allowed for new forms of control and agency on a general level. Overall, considering that faculty may have lost but also gained some autonomy during the onset of the COVID-19 crisis, we had no reason to assume a reduced level of satisfaction of the need for autonomy among faculty teaching during the COVID-19 crisis as compared to those teaching before.

The authors are encouraged to move the sentence in line 154 and line 155 to the end of that paragraph.

Author Response: Thank you for this further constructive and concise suggestion to improve readability and structure of our text. In response, the sentence was deleted and integrated towards the end of the paragraph.

The revised text reads as follows:

Overall, we hypothesized that faculty teaching online during the COVID-19 crisis would perceive their relatedness with students being considerably impaired compared to those teaching face-to-face before and that this effect would be exacerbated, the fewer students would be visible to faculty during a synchronous, video-based online class.

Please move the following contents to the “SDT and faculty well-being” section:

There is initial empirical evidence that the satisfaction of one or more of the three basic psychological needs for autonomy, competence, and relatedness was associated with different aspects of subjective well-being, including higher levels of positive emotions, lower levels of negative emotions, and higher teaching satisfaction; this evidence stems from studies conducted both before and during the pandemic [15,27,29,37,38].

Author Response: We agree and have moved the information correspondingly. 

The revised text reads as follows:

Nevertheless, there is initial evidence from the higher education context that the satisfaction of one or multiple psychological needs was associated with intrinsic motivation, higher levels of positive emotions, lower levels of negative emotions, and higher (teaching) satisfaction [16,18,28–32].

Please divide the first paragraph of the Results into two or three paragraphs to improve the readability. 

Author Response: Again, thank you for this constructive suggestion, we created separate paragraphs for the sample comparison with respect to sample characteristics, the within-participant comparison with respect to working conditions, and the main sample comparison.

The authors should make sure they cite prior work adequately in the Discussion section.

Author Response: Thank you for bringing this to our attention. We double-checked the discussion section and added references whenever possible.

The revised text reads as follow 

Surprisingly, faculty teaching online during the pandemic compared to faculty teaching before the pandemic reported impaired satisfaction not only of the needs for competence and relatedness but also for autonomy. The reduction in relatedness between faculty teaching before and during the crisis was also supported by means of mental contrasting between in-crisis and pre-crisis teaching within the COVID-19-sample, but the reduction in perceived autonomy and competence were not.

We propose that this differential pattern of findings for competence and autonomy can be explained by the different measurement approaches. While the between-person sample comparison with other faculty teaching face-to-face before the pandemic compared in-situ experiences within teaching sessions, the within-person mental contrast involved faculty members’ more general beliefs about their own teaching experiences (see also [62], on discrepancies across state vs. trait self-reports of teaching emotions). For instance, with respect to perceived autonomy when teaching in general, faculty may have focused on their autonomy in choosing content, their preferred implementation of online teaching, or their gained flexibility [39,46,50] with respect to time and location, rather than on session-specific limitations, such as fewer teaching methods, when contrasting their current against previous experiences. With respect to competence, in making such a general comparison faculty may have focused on stable aspects of their own competence that apply to both online and face-to-face teaching, such as content knowledge. When judging their in-situ experiences right after having delivered a class online during the pandemic, however, technical hassles and restrictions implied by the digital format may have been more salient and may therefore have undermined their in-situ experiences of competence. Although technical problems did not significantly predict the satisfaction of the need for competence in our study, very likely due to measurement-inherent problems because technical problems were assessed generally and not situation-specific, they still seem to be a promising factor in promoting or thwarting online teaching competence when assessed situation-specific, which also aligns with notions on faculty readiness to teach online [43,52]. 

The varying results depending on the measurement approach, that is judging against own experiences versus comparing two groups, show that it does probably not describe the whole picture when research solely relies on general retrospective judgements that are supposed to compare experiences during the highly exceptional situation of the COVID-19 crisis to experiences before the crisis and that such findings need to be interpreted with caution.

The fact that the physical distance between faculty and their students was so salient that faculty consistently experienced severely reduced levels of perceived relatedness, irrespective of the measurement approach, speaks to the robustness of this finding, which is further corroborated by similar findings from studies conducted during the COVID-19 crisis without a control group (e.g., [48,50]). We consider this one of the key findings of the present study and propose that before the COVID-19 pandemic, the importance of relatedness in the higher educational context may have been underestimated, because it develops rather easily in face-to-face settings when regularly interacting with and getting to know each other [63,64]. Although there are possibilities to form relationships with students in online contexts, for instance by self-disclosure, that is revealing personal information, responding in a timely manner, and using humor [65,66], such offers probably cannot fully compensate the loss of recurrent classroom interactions that more naturally allow to develop mutual relationships between individual students and faculty members [63,64].

The authors should ensure that they do not present results in the Discussion section. For instance, contents from line 337 to line 342 should be moved to Result section.

Author Response: Thank you very much for pointing out that the presentation of the results in the discussion was too long. To address this issue and at the same time to be able to refer to the findings, we kept only a short reiteration of the main findings in the discussion section and moved it to the paragraphs where the respective findings were discussed. This ensures that there is no lengthy repetition of pure findings in the beginning.

The authors are encouraged to shorten “Limitations and directions for future research and higher education” section.

Author Response: Thank you for the suggestion. In response, we shortened that section, and we moved some of its content to other parts of the discussion. In addition, we created a new section “Implications for post-pandemic online teaching”, which actually represents the content better.

The manuscript may benefit from subdividing the Discussion into topics that the authors wish to discuss.

Author Response: Thank you for pointing this out. We tried to create more meaningful sections (see also prior comment).

There are many brackets in the manuscript. The authors are encouraged to reduce the use of brackets to improve the readability.

Author Response: Thank you for bringing our attention to this stylistic issue. We have gone through the manuscript and replaced as many parentheses as possible. One section left with many parentheses is the “Results” section. Here, however, the parentheses ensure that the effect sizes and evidence as indicated by the BFs is reported accurately and as concisely as possible without making the paragraph very lengthy. Therefore, we retained those parentheses as well.

---

## [Decision Letter · Decision Letter 1]

26 Jul 2022

“I’m tired of black boxes!”: A systematic comparison of faculty well-being and need satisfaction before and during the COVID-19 crisis

PONE-D-22-04848R1

Dear Dr. Schwab,

We’re pleased to inform you that your manuscript has been judged scientifically suitable for publication and will be formally accepted for publication once it meets all outstanding technical requirements.

Kind regards,

Heng Luo, Ph.D.

Academic Editor

PLOS ONE

Additional Editor Comments (optional):

Dear Dr. Schwab, I think you have done a good job addressing all the reviewer comments in the first round of review. One reviewer is completely satisfied with the quality of your revision. The other reviewer, despite the unfavorable rating, raised several additional issues that I think are minor and revisable. Regarding the innovation of the article as pointed out by the second reviewer, since Plos One clearly indicates that innovation shouldn't be the sole deciding factor for article acceptance, I will skip this review comment. Please go through the manuscript one more time and conform it to the Plos One format for publication. Thank you, and congratulations!

Reviewers' comments:

Reviewer's Responses to Questions

**Comments to the Author**

1. If the authors have adequately addressed your comments raised in a previous round of review and you feel that this manuscript is now acceptable for publication, you may indicate that here to bypass the “Comments to the Author” section, enter your conflict of interest statement in the “Confidential to Editor” section, and submit your "Accept" recommendation.

Reviewer #1: (No Response)

Reviewer #2: All comments have been addressed

2. Is the manuscript technically sound, and do the data support the conclusions?

Reviewer #1: Yes

Reviewer #2: Yes

3. Has the statistical analysis been performed appropriately and rigorously? 

Reviewer #1: Yes

Reviewer #2: Yes

4. Have the authors made all data underlying the findings in their manuscript fully available?

Reviewer #1: Yes

Reviewer #2: Yes

5. Is the manuscript presented in an intelligible fashion and written in standard English?

Reviewer #1: Yes

Reviewer #2: Yes

6. Review Comments to the Author

Reviewer #1: The revised manuscript has been improved according to the reviewers’ suggestions, such as, applying two approaches to compare face-to-face vs. online teaching and strengthening the demonstration of the relationships among research variables. However, the authors are suggested to consider the following improvement.

First, it is suggested that the author should systematically consider the all captions to form a clear paper level caption, so as to make the paper more logical and readable. For example, illustrating whether the chapter of Procedure and measures, Sample and Statistical analyses are subordinate to the chapter of Method.

Second, the study took two approaches to investigate the comparisons of face-to-face teaching and online teaching, including internal retrospective comparison and group comparison. It is necessary to supplement it in the Abstract.

Third, in this study, teaching format was included in quantitative analysis as a research variable. It should be explained what are the features of the concept of teaching format and how to measure it. As far as I am considered, teaching format means a context that combined multiples and interconnected factors. It seems difficult to quantify the teaching format. The authors can consider discussing the relationships between basic needs and subjective well-being in online and offline teaching contexts respectively, instead of taking the teaching format as a specific research variable. In addition, there are many studies on the relationships between basic needs and subjective well-being, so that this study is lack of innovation.

Reviewer #2: This article has been carefully revised in the light of the reviewers' comments. The revised version can contribute to the theory and practice of SDT and teacher wellbeing.

7. PLOS authors have the option to publish the peer review history of their article (what does this mean?). If published, this will include your full peer review and any attached files.

Reviewer #1: No

Reviewer #2: No

---

## [Editor Report · Acceptance letter]

15 Sep 2022

PONE-D-22-04848R1 

“I’m tired of black boxes!”: A systematic comparison of faculty well-being and need satisfaction before and during the COVID-19 crisis 

Dear Dr. Schwab:

I'm pleased to inform you that your manuscript has been deemed suitable for publication in PLOS ONE. Congratulations! Your manuscript is now with our production department. 

Kind regards, 

on behalf of

Dr. Heng Luo 

Academic Editor

PLOS ONE